# Effects of Strength Training on Body Fat in Children and Adolescents with Overweight and Obesity: A Systematic Review with Meta-Analysis

**DOI:** 10.3390/children9070995

**Published:** 2022-07-01

**Authors:** Luis Diego Méndez-Hernández, Esther Ramírez-Moreno, Rosario Barrera-Gálvez, María del Consuelo Cabrera-Morales, Josefina Reynoso-Vázquez, Olga Rocío Flores-Chávez, Lizbeth Morales-Castillejos, Nelly del Socorro Cruz-Cansino, Reyna Cristina Jiménez-Sánchez, José Arias-Rico

**Affiliations:** 1Health Public Master, Academic Area of Medicine, Dr. Eliseo Ramirez Ulloa 4000, Pachuca 42090, Hidalgo, Mexico; me450805@uaeh.edu.mx (L.D.M.-H.); maria_cabrera10370@uaeh.edu.mx (M.d.C.C.-M.); 2Academic Area of Nutrition, Interdisciplinary Research Center, Circuit Actopan Tilcuautla s/n. Ex Hacienda La Concepción, Pachuca 42160, Hidalgo, Mexico; esther_ramirez@uaeh.edu.mx (E.R.-M.); ncruz@uaeh.edu.mx (N.d.S.C.-C.); 3Academic Area of Nursing, Institute of Health Sciences, Circuit Actopan Tilcuautla s/n. Ex Hacienda La Concepción, Pachuca 42160, Hidalgo, Mexico; rosario_barrera@uaeh.edu.mx (R.B.-G.); ofloresc@uaeh.edu.mx (O.R.F.-C.); lizbeth_morales@uaeh.edu.mx (L.M.-C.); cristyji@hotmail.com (R.C.J.-S.); 4Academic Area of Farmacy, Interdisciplinary Research Center Institute of Health Sciences, Circuit Actopan Tilcuautla s/n. Ex Hacienda La Concepción, Pachuca 42160, Hidalgo, Mexico; jreynoso@uaeh.edu.mx

**Keywords:** childhood obesity, strength training, body fat

## Abstract

Childhood overweight and obesity represent a growing public health problem worldwide. Since the 1980s, the global prevalence of overweight and obesity in childhood and adolescence has increased by 47%. The promotion of exercise is an important intervention to reduce the physical damage of obesity. The meta-analysis was conducted in accordance with the general guidelines for the reporting of systematic reviews and meta-analyses (PRISMA). The PubMed, SciELO, ScienceDirect and Google Scholar databases were searched from August to December 2021. The search yielded 722 titles published between 2000 and 2021. After screening the titles and abstracts, 64 duplicate articles were detected, and 27 articles were ultimately included in the systematic review, including 26 articles published in English and one published in Spanish. There was a statistically significant effect of the strength training interventions on the percentage of body fat, Test of 0 i = (*p* = 0.00, z = 6.92), Test of 0 = (*p* = 0.00, Q (9) = 42.63). The findings reveal that strength training has a positive impact on the treatment of body fat in children and adolescents with overweight and obesity.

## 1. Introduction

Childhood overweight and obesity is a global public health problem [1]. It is defined as a condition in which the body carries excessive and unhealthy amounts of body fat, which causes the individual to weigh more than their ideal weight [2]. This complex and multifactorial disease can begin in childhood and may be caused by a genetic-environmental interaction [3].

An imbalance between the energy expended (exercise) and the energy ingested (food) could lead to overweight and obesity. When food intake regularly exceeds calorie expenditure, unused energy is stored in adipose tissue or body fat [4]. Most people who eat and drink more than they use energy will produce adipose tissue to store excess energy [5]. There are many children who have a normal BMI but have a high percentage of body fat and low muscle mass. The promotion of exercise is an important intervention to reduce the physical damage of obesity [6,7]. 

Since the 1980s, the global prevalence of overweight and obesity in childhood and adolescence has increased by 47%. This trend has been observed in developed countries (where 24% of boys and 23% of girls are overweight or obese) and developing countries (where 13% of boys and girls are overweight or obese). Although the prevalence of childhood overweight and obesity in developed countries may have begun to stabilize, it continues to increase in developing countries, while socioeconomic inequalities in the prevalence of overweight and obesity persist or are expanding in many populations [1]. 

Childhood obesity has many harmful effects and comorbidities (i.e., associated diseases and disorders) that frequently produce a metabolic syndrome that greatly increases the risk of the individual remaining in an obesogenic state in adolescence and adulthood; 50% of the children who are obese at 6 years remain obese throughout adulthood, and 80% of the children who are obese in adolescence remain obese throughout adulthood [8]. Obesity has serious effects on life expectancy and quality of life, including hyperinsulinemia (high insulin levels), hypertension (high blood pressure) and dyslipidemia (abnormal lipid levels), which can cause heart disease, diabetes, cancer, hypertension and kidney diseases [9]. 

Strength Training (ST) is a term used to denote a component of sport and physical training that is designed to improve muscle strength, muscle power and muscular endurance, with a wide range of resistive loads, from body weight to weight [10,11]. This type of program can include the use of free weights (bars and dumbbells), weight machines, medicine balls, kettlebells, elastic tubes or a person’s own body weight [12,13]. 

In the past, strength training was not advisable before the end of sexual maturation due to the lack of hormonal status, especially of testosterone, typical of children and adolescents. It was suspected that strength training caused a negative effect on growth and bone maturation, consequently increasing the predisposition to trauma, especially with respect to the epiphyses, growth cartilages, bones, and bone connective tissue [11,13]. 

Before puberty and after the age of 70, strength training is not useful from physiological perspective [12]. It is currently possible to verify that the human organism is trainable throughout the life period, although this possibility of training is subject to notable changes that depend on the individual development phase [14]. 

A meta-analysis conducted in 2018 analyzed the effects of strength training on weight status in children and adolescents with normal weight, overweight and obesity across 24 randomized controlled trials [15]. After examining body mass (kg), BMI, body fat (%), lean mass (kg), skinfolds (mm) and waist circumference (cm), the analysis revealed a significant effect of Strength Training (ST) on body fat, fat-free mass, fat mass and waist circumference but no significant overall effect on body mass and BMI. These findings indicate that ST could play a role in the treatment and prevention of obesity.

However, another meta-analysis conducted in 2013 [16] identified that strength training in children and adolescents with overweight and obesity seems to have very small-to-small effects on body composition and moderate-to-large effects on strength.

The benefits of ST in children and adolescents have been documented, and organizations such as NSCA, UKSCA and AAP (American Academy of Pediatrics) have developed statements in a position of support [10,17,18]. One of these benefits is the positive effect of ST on weight status by reducing body fat, increasing muscle mass and improving the function of nonskeletal muscle tissues; however, the underlying molecular mechanisms of these beneficial effects are largely unknown [19]. 

Most research on the benefits of exercise on adipose tissue has focused on aerobic exercise [19]. Thus, there is a lack of information on the systemic benefits for body composition of strength exercise beyond improvements in muscle function.

The objective of this study was to perform a systematic meta-analysis to determine the effect of strength training on body fat in children and/or adolescents with overweight or obesity.

## 2. Materials and Methods

### 2.1. Study Search and Selection Strategies

The present meta-analysis was performed in accordance with the general guidelines for the reporting of systematic reviews and meta-analyses (PRISMA) See Figure 1. The PubMed, SciELO, ScienceDirect and Google Scholar databases were systematically searched from August to December 2021. The registration protocol was carried out in the Open Science Framework with the registration number 10.17605/OSF.IO/MNCNBI.

The following keywords were used in English and Spanish. Target population: Youth, youth, child, adolescent, puberty, boys, girls; Strength training: resistance training, muscle strength training, muscle strength program, muscle strength intervention, muscle strength exercise, weight training, strength and conditioning, concurrent training; Weight status: obese, overweight, body composition, waist circumference, fat, body mass, weight, skin fold.

For studies that did not provide access to the complete document or studies that lacked the necessary descriptive statistics, the authors were contacted via email.

### 2.2. Inclusion and Exclusion Criteria 

The inclusion criteria were as follows: (I) participants were aged between 5 and 19 years, (II) body fat was analyzed, (III) randomized controlled trials, (IV) a muscle strength training was implemented and (V) original publications in English or Spanish published from 2000 to 2021.

The exclusion criteria were as follows: (I) the subjects had a pathological condition or disability that affected movement (e.g., cerebral palsy/dyspraxia), (II) the subjects were found to have a behavioral or neuropsychological condition and (III) plyometric, vibratory or neuromuscular training was used, or training specifically for rehabilitation purposes was implemented.

### 2.3. Data Extraction 

The data were extracted using an electronic form and included study characteristics (country, year of publication, sample), characteristics of the participants (age, nutritional status), components of the intervention (duration, type of training), outcome measures (body fat, BMI, waist circumference) and methodological quality. The study was carried out by two researchers, they evaluated the selection of titles and abstracts. Following that, full articles will use the inclusion and exclusion criteria. No automated tools were used in the process.

### 2.4. Evaluation of Methodological Quality and Risk of Bias

To evaluate the methodological quality and the risk of bias of the included studies, the “Quality assessment tool for quantitative studies” developed by the public health practice project was used [20]. The results of the evaluation led to a general methodological rating of strong, moderate or weak in eight sections: selection bias, study design, confounding factors, blinding, data, collection methods, elimination and dropouts, originality of the treatment and analysis. This evaluation tool has been proven to be valid and reliable. To verify the reliability, this evaluation was carried out in 100% of the included studies, and any disagreement was resolved by discussion between the two authors.

### 2.5. Data Analysis

The reference data were expressed as numbers, proportions, averages and standard deviations. The meta-analysis was performed with the statistical package Stata/IC14.2 (Texas, USA), and a random effects model was used to examine the decrease or increase in percentage of body fat with 95% confidence intervals (CI). The T 2, H 2 and I 2 statistics were used to evaluate the heterogeneity of the studies; heterogeneity was not determined between the studies if I 2 was less than 50%, and an I 2 value equal to or greater than 50% indicated heterogeneity. A value of H2 equal to 1 means that there is no heterogeneity, and its value increases when heterogeneity between studies increases. To identify publication bias, the effect sample sizes were plotted against standard errors.

## 3. Results

### 3.1. Search Results

A total of 722 articles were identified from the PubMed, Science Direct, Scopus, SciELO and Google Scholar databases. In the screening phase, duplicates were eliminated, and the studies were filtered by screening the titles, abstracts and keywords, resulting in 361 references. A total of 169 studies were subjected to full-text analysis, with 55 studies being excluded because they were not focused on strength training, 49 being excluded because they did not specify the type of training applied, 36 being excluded because they did not evaluate variables related to body fat, and 2 being excluded because they did not specify the state anthropometric parameters of the children. Twenty-eight studies met the selection criteria [21,22,23,24,25,26,27,28,29,30,31,32,33,34,35,36,37,38,39,40,41,42,43,44,45,46,47,48].

### 3.2. Characteristics of the Studies

Across the 28 included articles, 1834 children and adolescents with overweight or obesity were analyzed, with an average age (years) of 12.5 ± 2.6 (min: 6 max: 19). The average duration (weeks) was 12 ±. 7.3, the average frequency (weekly) was of 3 ± 1.1 days, the average duration (minutes) of 60 ± 10.1 per session, and the average sets per exercise was 3 ± 1.3 with an average of repetitions per exercise of 12 ± 6.6. The studies were conducted in 12 countries: Brazil (10 studies), the United States (4 studies), Australia (3 studies), Canada (2 studies), France (2 studies), Chile (1 study), China (1 study), South Korea (1 study), Spain (1 study), Germany (1 study), Italy (1 study) and Austria (1 study). After the interventions in 92% of the studies, body fat was reduced, the remaining 8% did not present significant results, and the average attendance rate for the studies that measured this variable was 75%. For the studies that provided information, 15% presented a low intensity <50% maximum repetition (MR), 35% medium intensity 50–75% MR, and 50% a high intensity >75%.

### 3.3. Results of Methodological Quality

The Table 1 show the results of the methodological quality assessment, where the studies obtained a moderate score of 60.7%, a strong score of 35.7% and a weak score of 3.6%.

### 3.4. Results of the Meta-Analysis

Ten studies reported the necessary statistics to construct the forest plot and funnel plot [21,25,31,32,37,39,44,46,47,48]. Using the random effects model, the results of the percentage of body fat were compared after the interventions with 95% CIs. The results of the meta-analysis revealed that ST is an effective intervention for the treatment of body fat (*p* = 0.00, z = 6.92), (*p* = 0.00, Q (9) = 42.63). The heterogeneity parameters of T 2 = 1.69, H 2 = 3.56 and I 2 = 71.94% indicated significant heterogeneity. See Figure 2 and Figure 3.

### 3.5. Subgroup Analysis

In an attempt to determine the causes of heterogeneity in our analysis, subgroup analyzes were performed according to the intensity of the studies (A: High and medium intensities >51% MR, B: Low intensities <50% MR). See Figure 4.

(A) Five studies were used to construct medium and high intensity subgroup analyzes [25,34,36,41,42], showing that medium and high intensity ST could be an effective intervention for the treatment of body fat loss (*p* = 0.03, z = 2.14), (*p* = 0.02, Q (4) = 11.49). (B). Therefore, the studies to construct low intensity subgroup [21,26,30], were not statistically significant for the treatment of body fat (*p* = 0.14, z = 1.47), (*p* = 0.19, Q (2) = 3.30).

## 4. Discussion

According with the position statements of the American Academy of Pediatrics [10], UKSCA [17] and NSCA [18] on youth strength training suggest that it can have a positive impact on body fat, and the significant findings of this meta-analysis for body fat percentage are conclusive, supporting these claims. Regardless of dietary restrictions [50], being more effective at training in the gym [22,24,35,41,42] than home workouts [26,31]. Strength training showed a statistically significant effect on body fat percentage. Although this decrease in fat is expected, it is necessary to investigate in-depth studies considering the individual response to the intensity of strength training. 

The training intensities that presented the best benefits for the reduction of body fat were high and medium [22,24,34,42,45], a possible cause may be that these intensities are close to muscle failure > 5 repetitions in reserve, this has been shown to be favorable [14], additionally if it is added to an adequate diet [22,37,51] these results are increased.

In the funnel plot (Figure 3), an asymmetry can be observed, which indicates publication bias. A possible cause is that due to the very specific research topic, there are a limited number of relevant studies; likewise, the existing studies have low sample sizes because the inclusion criteria are only children or adolescents with overweight or obesity.

It has been suggested that training periods of more than 14 weeks are required to observe effects on body fat and increases in lean muscle mass [43]. This suggests that the duration of the intervention for several of the studies may not have been long enough to invoke positive measurable changes.

Although more studies are required to provide a better understanding of the mechanism of a reduction in body fat due to a strength training intervention, it has been reported that a possible cause would be that the ST itself favors the loss of fat through the muscle. The muscle tissue releases Extracellular Vesicles (EVs) that present a potential mechanism through which the beneficial effects of exercise are transmitted to other tissues [52,53]. These EVs contain miR-1 absorbed by adipose tissue, promoting adrenergic signaling and lipolysis in adipose tissue, which favors fat oxidation [19]. Alternatively, it may simply be due to an increase in skeletal muscle mass and the resulting increase in basal metabolic rate [54]. In particular, this has been observed in adolescents [55]. 

On the other hand, it is also suggested that these changes could be due to increases in total energy expenditure that may have occurred simply when participating in an exercise intervention instead of an increase in metabolically active lean tissue. However, it is important to note that the data of the participants, children and adolescents included in the analysis may have had an impact on the results.

There are limitations in this study, that must be taken present into account when explain the results. First, the included studies used several different types of exercise intervention (exclusive strength training or strength training plus aerobic training) with great variability within the study interventions with respect to the number of participants (ranging from 18 to 304 participants). The duration varied from 6 to 52 weeks, the frequency varied from 2 to 5 times per week, and the programs also included a mixture of a series of 1 to 4 per exercise and repetitions with a range of 4 to 25 with low intensities, medium or high according to the 1MR, with different methodologies such as undulating periodization, training of a progressive nature with an increase in the number of sets, repetitions and resistance; in circuit, with linear periodization or with daily undulating periodization, circuit training; and with training at home or in the gym, added to the fact that the age of entry was not measured.

For the results, there were a variety of different measurement methods; for example, the percentage of body fat was measured by DEXA, BodPod, bioelectrical impedance, skinfolds and magnetic resonance. In the analysis, not all studies reported data to allow an exhaustive investigation, so limited conclusions can be drawn based on this level of additional analysis.

## 5. Conclusions

The results of this systematic review confirm that ST could be an effective intervention for the treatment of body fat percentage in the first 14 weeks of intervention with better long-term results (>36 weeks), in turn, high and medium intensities are beneficial for reducing body fat percentage. However, a more in-depth research is needed on ST intensities and their effect at the individual level in children and adolescents, these findings can be used to develop new methods for the treatment of childhood obesity.

## Figures and Tables

**Figure 1 children-09-00995-f001:**
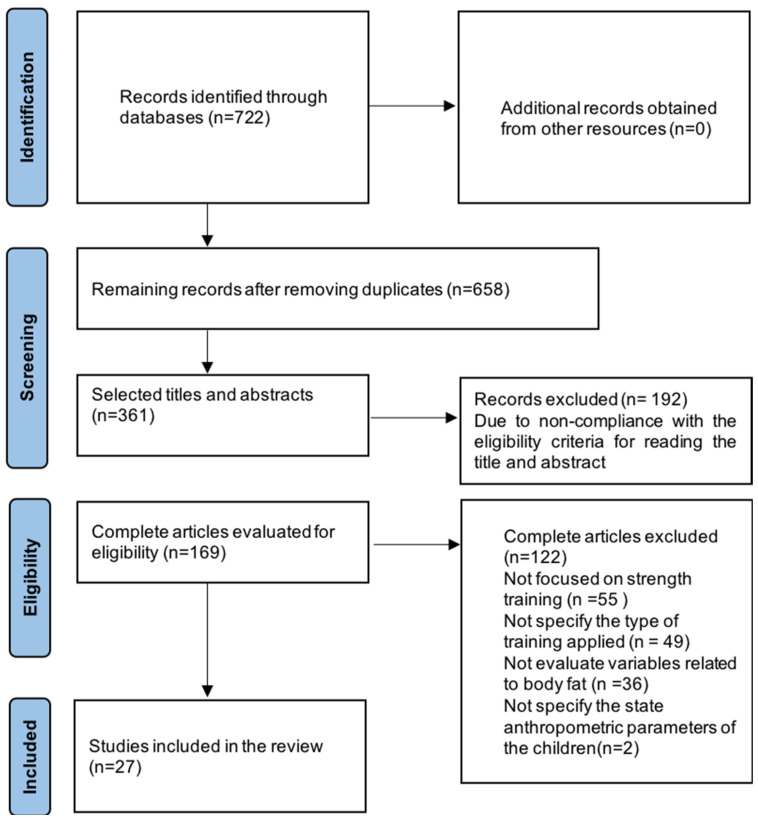
PRISMA flowchart of systematic search and included studies.

**Figure 2 children-09-00995-f002:**
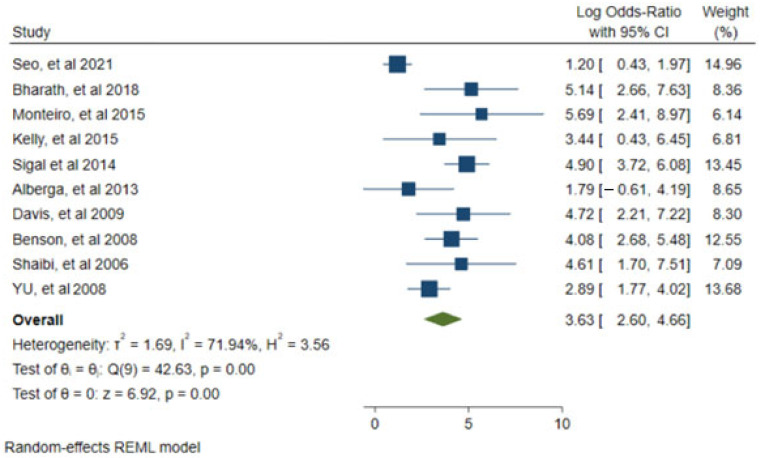
Forest plot of the random effects model for the meta-analysis between the correlation of reduction or increase in percentage of body fat. Data from: [21,25,31,32,37,39,44,46,47,48].

**Figure 3 children-09-00995-f003:**
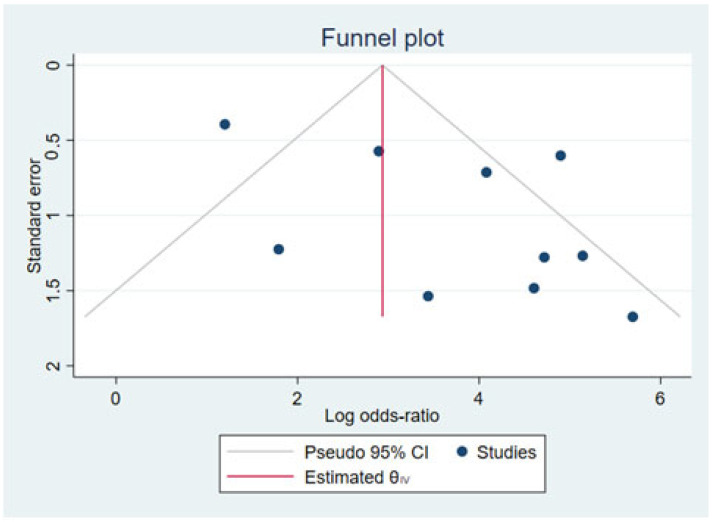
Funnel plot using sample sizes of the effect against standard errors Data from: [25,31,32,37,44,46,47,49].

**Figure 4 children-09-00995-f004:**
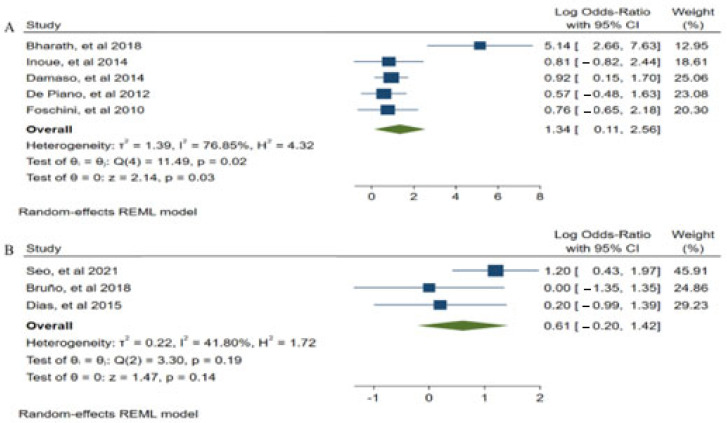
Subgroup analysis ((**A**): High and medium intensities > 51% MR, (**B**): Low intensities < 50% MR) Data from: [21,25,26,30,34,35,41,42].

**Table 1 children-09-00995-t001:** Quality assessment for quantitative studies.

Folio	Study	Country	Sample	Age	Nutritional Status	Duration	Type of Training	Outcome Measures	Methodological Quality
1	Seo, Y. G et al., 2021 [21]	South Korea	A quasi-experimental intervention trial including 242 participants. Participants were categorized into three groups to receive usual care (usual care group), exercise intervention with circuit training (exercise group) or intensive nutritional intervention and feedback with a balanced diet (nutritional group).	Between 6 and 17 years. Average age: 11	Overweight and obesity	Phases 1, 2, 3 and 4 correspond to the intensive intervention (for 6 months), group activity (for 6 months), reinforcement (for 3 months) and group activity (for 9 months), respectively. The intervention lasted for 24 weeks, 3 days a week for 60 min per session (one group exercise session and two home exercise sessions) (72 sessions in total)	(RT) Full body, plus aerobic exercise (home exercises)	Body fat (%), fat-free mass (kg), BMI.	3
2	Miguet, M. et al., 2020 [22]	France	A total of 43 adolescents were divided into two groups: MICT (*n* = 21) or HIIT (*n* = 22).	Between 11 and 15 years. Average age: 13	Obesity	The intervention lasted 16 weeks: 4 sessions per week, nutritional education (2/month) and psychological support (1/month) (64 sessions in total).	Aerobic: HIIT and MICT plus (RT) Whole body	Body fat (%), fat-free mass (kg), BMI.	2
3	B. Horsak et al., 2019 [23]	Austria	A total of 51 children and adolescents divided into 2 groups (experimental group = 26/control Group *n* = 25).	Between 10 and 18 years. Average age: 13	Obesity	The intervention lasted 12 weeks: 2 sessions per week, duration 60 min each session (24 sessions in total).	(RT) Lower body	Body fat (%), fat-free mass (kg)	2
4	Branco, B. H. M, et al., 2018 [24]	Brazil	A total of 18 adolescent males divided into 2 groups (Weightlifting = 9/functional training = 9)	Between 15 and 17 years. Average age: 16	Obesity	The intervention lasted 12 weeks. From 1 to 6 weeks: 3 sessions per week, duration 46 min each session. From 7 to 12 weeks: 3 sessions per week, 52 min each session (36 sessions in total).	(RT) Whole body	Body fat (%), fat-free mass (kg), BMI, WC, WC	1
5	Bharath, L et al., 2018 [25]	Germany	A total of 40 adolescents were randomly assigned to a group of exercise training (EX; *n* = 20) or without exercise (CON; *n* = 20).	Average 14	Obesity	The CRAE program was performed for 60 min, 5 times a week for 12 weeks. (60 sessions in total)	(RT) Whole body plus aerobic exercise	Body fat (%), fat-free mass (kg), BMI, WC.	2
6	Bruñó, A et al., 2018 [26]	Spain.	A total of 52 participants were randomly assigned to a print (*n* = 18), web-based (Move It; *n* = 18) or Move It plus support group (*n* = 16).	Between 9 and 16 years. Average age: 12	Obesity	The intervention included 60 sessions distributed over 12 weeks, with five weekly sessions of 60 min each.	(RT) Full body, plus aerobic exercise (home exercises)	Body fat (%).	3
7	Fiorilli, G et al., 2017 [27]	Italy	A total of 41 overweight adolescents were randomly divided into three exercise groups: in the first phase, the first group performed a 16-week moderate-intensity strength training (RT), the second group performed a 16-week high-intensity RT, and the third group performed aerobic training (AT) for 16 weeks. In the second phase, all groups performed aerobic exercise for 6 weeks.	Between 12 and 15 years.	Obesity	The intervention lasted 22 weeks: 3 weekly sessions, 40 min per session.	(RT) Medium intensity and high intensity full body and aerobic exercise	Body fat (%), fat-free mass (kg), BMI, WC.	2
8	Rey, O et al., 2017 [28]	France	A total of 24 remaining participants (14 girls and 10 boys).	Between 14 and 15 years.	Obesity	The intervention lasted 5 weeks: 3 sessions per week, 45 min sessions.	Vigorous aerobic training	Body fat (%), fat-free mass (kg), BMI.	2
9	Crouter, S. et al., 2016 [29]	United States	A total of 30 participants.	Between 7 and 18 years. Average age: 11	Obesity	The intervention lasted 24 weeks: 4 times a week for a duration of 90 min. It included aerobic, strength and self-organized exercises.	(RT) Whole body and aerobic exercise	Body fat (%), fat-free mass (kg), BMI.	2
10	Dias, I, et al., 2015 [30]	Brazil	A total of 24 adolescent women with obesity composed the experimental group. The control group included 20 adolescents without obesity.	Between 13 and 17 years. Average 14	Obesity	The intervention lasted 12 weeks: 3 sessions per week on nonconsecutive days, duration of 30–40 min each session (36 sessions in total).	(RT) Whole body	Body fat (%), fat-free mass (kg), BMI	2
11	Kelly, L. A, et al., 2015 [31]	United States	A total of 26 Latino adolescents, randomly assigned to a strength training group (*n* = 13) or a control group (*n* = 13).	Between 14 and 18 years. Average 15	Obesity	The intervention lasted 16 weeks, 2 weekly sessions, duration of 1 h post-session (32 sessions in total).	(RT) Whole body The HBST intervention was performed in the homes of the participants.	Body fat (%), fat-free mass (kg), BMI	2
12	Monteiro, P et al., 2015 [32]	Brazil	A total of 32 adolescents with obesity participated in two randomized training groups, concurrent (*n* = 14) or aerobic (*n* = 18), and were compared with a control group of (*n* = 16).	Between 11 and 17 years. Average age: 11	Obesity	The intervention lasted 20 weeks (50 min x 3 per week, supervised).	Concurrent training and aerobic training	Body fat (%), fat-free mass (kg), BMI, WC.	2
13	Antunes, B et al., 2015 [33]	Brazil	A total of 25 adolescents with obesity.	Between 12 and 15 years. Average age: 13	Obesity	The intervention lasted 20 weeks, 3 times a week, with a duration of 60 min per session.	Concurrent training	Body fat (%), fat-free mass (kg), BMI, TG, CL, SBP, DBP.	2
14	Inoue, D et al., 2015 [34]	Brazil	A total of 45 adolescents were randomly assigned to three groups: aerobic training (EA *n* = 20), aerobic training plus strength with linear periodization (LP *n* = 13) and aerobic training plus strength training with daily undulating periodization (DUP *n* = 12).	Between 15 and 18 years. Average age: 16	Obesity	The intervention lasted 26 weeks: 3 times a week, with 60-min sessions.	Aerobic training, (RT) with linear periodization and (RT) with daily undulating periodization.	Body fat (%), fat-free mass (kg), BMI, TG, CL.	2
15	Dâmaso, A et al., 2014 [35]	Brazil	A total of 116 adolescents were randomly assigned into two groups: AT (*n* = 55), AT + RT (*n* = 61). 82 girls and 57 boys	Between 15 and 19 years. Average age: 16	Obesity	The intervention lasted 52 weeks (1 year): three times a week, including 60 min per session. (30 min of AT and 30 min of strength training).	(RT) Whole body and aerobic exercise	Body fat (%), fat-free mass (kg), BMI, Glc, Ins, HOMA-IR, TG, Leptin, Adiponectin.	3
16	Ackel-D’Elia et al., 2014 [36]	Brazil	A total of 72 adolescents, 50 girls and 22 boys The subjects were randomized into 3 groups: 1) aerobic training plus resistance (AT + RT: *n* = 24; 9 boys and 15 girls); 2) aerobic training (AT: *n* = 24; 7 boys and 17 girls) and 3) leisure physical activity (LPA: *n* = 24; 6 boys and 18 girls).	Between 15 and 19 years. Average age:	Obesity	The intervention lasted 24 weeks: 60 min per session and performed 3 times per week.	(RT) Whole body and aerobic exercise	Body fat (%), fat-free mass (kg), BMI, Glc, Ins, HOMA-IR, Leptin.	3
17	Sigal et al., 2014 [37]	Canada	A total of 304 participants were randomized to the following 4 groups for 22 weeks: aerobic training (*n* = 75), strength training (*n* = 78), combined aerobic and resistance training (*n* = 75) or control without exercise (*n* = 76).	Between 14 and 18 years. Average age: 15	Obesity	The intervention lasted 24 weeks: 4 days a week, 50 min per session.	(RT) Whole body and aerobic exercise	Body fat (%), fat-free mass (kg), BMI, WC, SBP, DBP, Glc, Ins, CL, TG.	3
18	Vásquez, F, et al., 2013 [38]	Chile	A total of 111 children were included. Early intervention group = 60 executed in parallel physical exercise of muscular strength, food education and psychological support for 3 months. Late intervention group = 51 incorporated the first 3 months, only educational intervention and psychological support, and exercise was added between 3 and 6 months.	Between 8 and 13 years.	Obesity	The intervention lasted three months (12 weeks): three times a week on nonconsecutive days, with a duration of 45 min (36 sessions in total).	(RT) Whole body	BMI, fat mass	3
19	Alberga, A. S, et al., 2013 [39]	Canada	A total of 19 children divided into 2 groups (moderate intensity strength exercises and high repetition (*n* = 12) and control group without intervention (*n* = 7).	Between 8 and 12 years. Average age: 10	Obesity	The intervention lasted 12 weeks: 2 sessions per week, duration of 75 min per session (24 in total).	(RT) Whole body	Body fat (%), fat-free mass (kg), BMI.	2
20	Antunes et al., 2013 [40]	Brazil	A total of 34 adolescents with obesity, including 12 girls and 22 boys in a single training group	Between 12 and 15 years. Average age: 13	Obesity	The intervention lasted 20 weeks: 3 times a week, with a duration of 60 min per session.	(RT) Whole body and aerobic exercise	Body fat (%), fat-free mass (kg), BMI, TG, CL.	2
21	De Piano et al., 2012 [41]	Brazil	A total of 58 adolescents (37 boys 32 girls) divided into two groups Aerobic (AT) *n* = 29, and Aerobic plus strength (AE + RT) *n* = 29.	Between 15 and 19 years. Average age: 16	Obesity	The intervention lasted 52 weeks: 3 times per week, 60 min per session.	(RT) Whole body and aerobic exercise	Body fat (%), fat-free mass (kg), BMI, HOMA-IR, CL, TG, Adiponectin, Leptin.	3
22	Foschini et al., 2010 [42]	Brazil	A total of 32 adolescents with obesity divided into two groups: linear periodization (LP) (*n* = 16) and daily undulating periodization (DUP) (*n* = 16).	Average age: 16 years	Obesity	The intervention lasted 14 weeks: three weekly sessions, 60 min per session.	(RT) Whole body	Body fat (%), fat-free mass (kg), BMI, Glc, Ins, HOMA-IR, CL, SBP, DBP.	2
23	Mc Guigan, M, et al., 2009 [43]	Australia	A total of 48 overweight and obese children (*n* = 26 girls and 22 boys; mean age = 9.7 years).	Between 7 and 12 years. Average age: 9	Overweight and obesity	The intervention lasted 8 weeks:, 3 weekly sessions (24 sessions in total).	(RT) Wavy periodized full body	Body fat (%), fat-free mass (kg), BMI.	2
24	Davis, J. N, et al., 2009 [44]	United States	A total of 54 overweight Latino adolescents (15.5 ± 1.0 years) were randomly assigned to: (i) Control (C; *n* = 16), (ii) Nutrition (N; *n* = 21), or (iii) Nutrition + Strength training. (N + ST; *n* = 17).	Between 14 and 18 years. Average age: 15	Overweight	In a 16-week randomized trial, 2 sessions per week on two nonconsecutive days (32 sessions in total), lasting 60 min each session.	(RT) Whole body	Total fat (kg), fat-free mass (kg), BMI.	3
25	Sgro, M et al., 2009 [45]	Australia	A total of 31 preadolescent children with overweight or obesity, divided into 3 groups. Group 8 (G8) trained for 8 weeks, Group 16 (G16) trained for 16 weeks, and Group 24 (G24) trained for 24 weeks.	Between 7 and 12 years. Average age: 9 years	Overweight and Obesity	Longitudinal design of 8, 16 and 24 weeks with 3 weekly sessions, between 45 and 60 min per session.	(RT) Full body with a combination of different power exercises and body weight with undulating variation (progressive)	Body fat (%), fat-free mass (kg).	2
26	Benson, A. C et al., 2008 [46]	Australia	A total of 78 children (32 girls and 46 boys). Divided into 2 groups: PRT intervention (37) and Control (41).	Average age: 12 years	Normal weight, Overweight and Obesity	The PRT group trained twice a week for 8 weeks (16 sessions in total).	(RT) Whole body	Body fat (%), fat-free mass (kg), BMI, WC.	3
27	Shaibi, G et al., 2006 [47]	United States	A total of 22 Latino adolescents were randomly assigned to a training group (RT = 11) or a control group without exercise (C = 11).	Average age: 15 years	Obesity	The intervention lasted 16 weeks: 2 weekly sessions of no more than 60 min (32 sessions in total).	(RT) Full body periodized and progressive in nature with an increase in the number of sets, repetitions and resistance used	Body fat (%), fat-free mass (kg), BMI.	2
28	YU, C et al., 2005 [48]	China	A total of 82 children from the Hong Kong school, divided into 2 groups (strength training with diet) or in the control group (diet only) (n 41 each).	Between 8 and 11 years. Average age: 10	Overweight and obesity	The training group attended a 75-min strength exercise program 3 times a week for 6 weeks (phase 1, 18 in total), after which it was offered and 22 children chose to continue with a program once per week for 28 more weeks with one session per week (phase 2, 46 sessions in total).	(RT) Complete body, in circuit	Body fat (%), fat-free mass (kg), BMI.	3

## Data Availability

Not applicable.

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
