# Peer review of "Effects of Strength Training on Body Fat in Children and Adolescents with Overweight and Obesity: A Systematic Review with Meta-Analysis"

_children, 2022, doi:10.3390/children9070995_

Round 1

Reviewer 1 Report

Dear authors! I would like to express my gratitude for the opportunity to get acquainted with such a large-scale work. As a discussion, I would like to know your opinion on the types of comorbidities referred to in the last sentence of section 5? What are these diseases?

Author Response

Response to reviewers

We would like to thank all reviewers for the comments they made to our manuscript entitled “Effects of strength training on body fat in children and adolescents with overweight and obesity: A Systematic Review”. Your contributions and suggestions have been considered and we include our responses point-by-point to the comments raised by the reviewers to improve the quality of the manuscript, including revision of the English language by a native speaker.

Reviewer 1.

Dear authors! I would like to express my gratitude for the opportunity to get acquainted with such a large-scale work. As a discussion, I would like to know your opinion on the types of comorbidities referred to in the last sentence of section 5? What are these diseases?

  • We highlight in the section of the introduction, line 63 that childhood obesity affects other conditions such as heart disease, diabetes, cancer, hypertension, and kidney diseases. Therefore, in the last sentence of section 5, we emphasize the non-communicable diseases related to obesity.

Reviewer 2 Report

First of all, the effect of strength or resistance training on body fat has been covered in previous studies and the results are relatively uniform, so it is not meaningful to discuss this paper as an entry point. Secondly, the research method of this paper is not rigorous enough. The intensity of strength training and intervention time may have different degrees of influence on individual body fat, and this paper does not perform a subgroup analysis on its division. Sensitivity analysis was not carried out for its related influencing factors. Finally, due to inappropriate entry points and imprecise research methods, the conclusions are too simple and the innovation is insufficient.

Author Response

We would like to thank all reviewers for the comments they made to our manuscript entitled “Effects of strength training on body fat in children and adolescents with overweight and obesity: A Systematic Review”. Your contributions and suggestions have been considered and we include our responses point-by-point to the comments raised by the reviewers to improve the quality of the manuscript, including revision of the English language by a native speaker.

Reviewer 2.

First of all, the effect of strength or resistance training on body fat has been covered in previous studies and the results are relatively uniform, so it is not meaningful to discuss this paper as an entry point.

  • We agree with what the reviewer said, for that, we included a sentence about the importance of investigating in-depth studies considering the individual response to the intensity of strength training. line 210 to 212

Secondly, the research method of this paper is not rigorous enough.

  • We have used the PRISMA checklist, which includes the basic criteria for a systematic review. We had included an attachment here as a separate file. Even we included in the paper now (Line 135 to 138): “We have included that the study was carried out by two researchers, they evaluated the selection of titles and abstracts. Following that, full articles will use the inclusion and exclusion criteria. No automated tools were used in the process”.

Also, in the methodology, we include some changes as:

  • (Line 152). For the evaluation of certainty (or confidence), it was included information about the confidence interval (95%) and the T 2, H 2, and I 2 statistics were used to evaluating the heterogeneity of the studies.

The intensity of strength training and intervention time may have different degrees of influence on individual body fat, and this paper does not perform subgroup analysis on its division. Sensitivity analysis was not carried out for its related influencing factors.

  • Line 181 to 183 We included “A single study does not provide information on training intensity, the remaining 26, 15% presented a low intensity <50% of maximum repetition (MR), 35% medium intensity 50-75% MR, and 50% a high intensity >75% MR.” and line 213 to 216

Finally, due to inappropriate entry points and imprecise research methods, the conclusions are too simple and the innovation is insufficient.

  • We modified the paper improving the introduction, methodology, results, and conclusions according to the reviewer´s recommendations. Therefore, the conclusion was modified in Line 258 to 263. The results of this systematic review confirm that ST could be an effective intervention for the treatment of body fat percentage in the first 14 weeks of intervention with better long-term results (> 36 weeks), in turn, high and medium intensities are beneficial for reducing body fat percentage. However, more in-depth research is needed on ST intensities and their effect at the individual level in children and adolescents, these findings can be used to develop new methods for the treatment of childhood obesity.

Reviewer 3 Report

The manuscript is a systematic review of studies assessing the effect of strength exercise on changes in body fat in children and adolescents with excess body weight. Searching for effective methods of treating overweight and obesity in children and adolescents seems advisable, but should not be considered in isolation from other methods of therapy. Shouldn't the importance of the influence of the diet on the effects of training also be discussed? Especially that in some of the presented works, nutritional education was also used.

The following phrases are inappropriate:

·         "... on the treatment of body fat ..." - "fat tissue" is not treated, but excessive body weight;

·         "Contrary to the beliefs that exist about the harmful effects of exercise on the health of children and adolescents ..." - should not refer to such a statement in conclusions, because it suggests the presence of such an influence.

In the conclusions the following statement was used: "these findings on the regulation of adipose tissue metabolism by strength training can be used to develop new methods ..." suggests discussing the effects of strength exercise on fat metabolism, but the authors did not refer to such results from the works described.

Should it not be necessary to indicate in Figure 2 whether the presented changes in adipose tissue are expressed in kilograms or in percent?

Why were only 169 out of 361 publications selected for full-text analysis?

The abbreviation EFM is not explained in the paper.

Author Response

We would like to thank all reviewers for the comments they made to our manuscript entitled “Effects of strength training on body fat in children and adolescents with overweight and obesity: A Systematic Review”. Your contributions and suggestions have been considered and we include our responses point-by-point to the comments raised by the reviewers to improve the quality of the manuscript, including revision of the English language by a native speaker.

Reviewer 3.

The manuscript is a systematic review of studies assessing the effect of strength exercise on changes in body fat in children and adolescents with excess body weight. Searching for effective methods of treating overweight and obesity in children and adolescents seems advisable, but should not be considered in isolation from other methods of therapy. Shouldn't the importance of the influence of the diet on the effects of training also be discussed? Especially in some of the presented works, nutritional education was also used.

  • Nutritional education is important for an intervention to reduce body fat. We include that “a healthy diet in addition to the training increases the result in the health of children. line 213 a 216

The following phrases are inappropriate:

  • "... on the treatment of body fat ..." - "fat tissue" is not treated, but excessive body weight;
  • Line 40 We had included the correct phrase: “It is defined as a condition in which the body carries excessive and unhealthy amounts of body fat,”

Round 2

Reviewer 2 Report

I am referring to the subgroup analysis in the forest plot or other data processing (not reflected in this article), rather than adding the words medium and high intensity. If you don't do subgroup analysis, how can you get the conclusion High-intensity strength training is good for fat loss. In the forest plot results in this paper, I2 is more than 50%, showing significant heterogeneity, and the influencing factors of strength training intensity were simply listed before. Whether other factors such as entry age and literature quality will have an impact on the results is also needed. For further analysis, it is said that the method is not scientifically rigorous enough.

Author Response

We would like to thank all reviewers for the comments they made to our manuscript entitled “Effects of strength training on body fat in children and adolescents with overweight and obesity: A Systematic Review”. Your contributions and suggestions have been considered and we include our responses point-by-point to the comments raised by the reviewers to improve the quality of the manuscript.

Reviewer 2.

I am referring to the subgroup analysis in the forest plot or other data processing (not reflected in this article), rather than adding the words medium and high intensity. If you don't do subgroup analysis, how can you get the conclusion High-intensity strength training is good for fat loss. In the forest plot results in this paper, I2 is more than 50%, showing significant heterogeneity, and the influencing factors of strength training intensity were simply listed before. Whether other factors such as entry age and literature quality will have an impact on the results is also needed. For further analysis, it is said that the method is not scientifically rigorous enough.

Line 217 a 229 we add.  

3.4. Subgroup analysis

In an attempt to determine the causes of heterogeneity in our analysis, subgroup analyzes were performed in Figure 4, according to the intensity of the studies (A: High and medium intensities >51% MR, B: Low intensities <50% MR)

  1. A) Five studies were analyzed to construct medium and high intensity subgroup analyzes [35,41,42,46,47], showing that medium and high intensity ST is an effective intervention for the treatment of body fat loss (p = 0.03, z = 2.14), (p = 0.02, Q (4) = 11.49). B). Therefore the studies analyzed to construct low intensity subgroup [23,33,36], were not statistically significant for the treatment of decreased body fat (p = 0.14, z = 1.47), (p = 0.19, Q (2) = 3.30).

Figure 4.  Subgroup analysis (A: High and medium intensities >51% MR, B: Low intensities <50% MR).

And we add “added to the fact that the age of entry was not measured”. Line 277-278, page 17 in the paragraph of discussion.

Round 3

Reviewer 2 Report

The author improves the article according to the revised comments to meet the publication requirements.